# Stability and In Vitro Aerodynamic Studies of Inhalation Powders Containing Ciprofloxacin Hydrochloride Applying Different DPI Capsule Types

**DOI:** 10.3390/pharmaceutics13050689

**Published:** 2021-05-11

**Authors:** Edit Benke, Patrícia Varga, Piroska Szabó-Révész, Rita Ambrus

**Affiliations:** Institute of Pharmaceutical Technology and Regulatory Affairs, University of Szeged, 6720 Szeged, Hungary; benke.edit@szte.hu (E.B.); varga.patricia@szte.hu (P.V.); ReveszPiroska@szte.hu (P.S.-R.)

**Keywords:** pulmonary drug delivery, powders for inhalation, dry powder inhaler, novel combined formulation, ciprofloxacin hydrochloride, sodium stearate, magnesium stearate, stability test, DPI capsules

## Abstract

In the case of capsule-based dry powder inhalation systems (DPIs), the selection of the appropriate capsule is important. The use of gelatin, gelatin-PEG, and HPMC capsules has become widespread in marketed capsule-based DPIs. We aimed to perform a stability test according to the ICH guideline in the above-mentioned three capsule types. The results of the novel combined formulated microcomposite were more favorable than those of the carrier-free formulation for all capsule types. The use of HPMC capsules results in the greatest stability and thus the best in vitro aerodynamic results for both DPI powders after six months. This can be explained by the fact that the residual solvent content (RSC) of the capsules differs. Under the applied conditions the RSC of the HPMC capsule decreased the least and remained within the optimal range, thus becoming less fragmented, which was reflected in the RSC, structure and morphology of the particles, as well as in the in vitro aerodynamic results (there was a difference of approximately 10% in the lung deposition results). During pharmaceutical dosage form developments, emphasis should be placed in the case of DPIs on determining which capsule type will be used for specific formulations.

## 1. Introduction

Research on pulmonary drug delivery (PDD) has been carried out in remarkable numbers in the last two and half decades, and the number of companies and research groups specializing in this field continues to grow [1]. This is due to the fact that the lung, as an alternative drug delivery gate, is able to absorb the drug over a large area according to its anatomical properties through a thin absorption membrane, and due to its excellent blood supply, a rapid systemic effect (much faster than oral administration) can be achieved [2]. Thus, PDD is suitable for both local and systemic therapeutic purposes [3]. Furthermore, it should be emphasized that it is much more advantageous compared to oral administration in terms of side effect profile, as the first-pass effect of the liver and the enzymatic inactivation of the gastrointestinal tract as metabolic pathways are avoided by the inhaled drug, requiring a lower therapeutic dose [4,5]. It is noteworthy that great emphasis is placed on the development of inhaled antibiotic products as, for example, they can be used effectively in the treatment of cystic fibrosis [6]. A number of inhaled antibiotics are currently available on the market, such as amikacin (Arikayce^®^, Insmed Incorporated, Bridgewater, NJ, USA), aztreonam (Cayston^®^, Cayston Gilead Sciences Ireland UC, Carrigtohill, Ireland), colistimethate sodium (Colobreathe^®^, Forest Laboratories UK Ltd., Whiddon Valley, UK), levofloxacin hemihydrate (Quinsair^®^, Chiesi Farmaceutici S.p.A., Parma, Italy), and tobramycin (TOBI^®^/TOBI^®^ Podhaler^®^, Novartis International AG, Basel, Switzerland; Bramitob^®^, Chiesi Farmaceutici S.p.A., Parma, Italy) [7,8]. In addition, many inhalation products containing antibiotics (e.g., ciprofloxacin, murepavadin, etc.) are in clinical trials [9,10].

For PDD, the following main groups can currently be distinguished: nebulizers, pressured metered-dose inhalers, soft mist inhalers and dry powder inhalers (DPIs) [11]. The development of the latter can be said to be the most popular of the listed, as their stability is relatively high compared to liquid-based systems due to solid powders, they are propellant-free to operate, easy to use, etc. [12,13]. For the optimal functioning of these microcomposites, in addition to the appropriate formulation, it is essential that patients use the inhalers professionally and master the correct breathing maneuver, and the development of DPI devices should facilitate the adequate flow of the formulation, must be compatible with the applied powder, however, that should allow easy application by the user [14]. A notable proportion of DPI products marketed are capsule-based [15], which suggests that remarkable attention should also be paid to the role of DPI capsules used, but the international literature has only recently begun to address this issue [16,17,18,19,20]

Capsules used in DPIs have different functions and properties compared to oral drug administration in terms of therapeutic success [21]. While capsules also play a role in the liberation of the drug when administered orally, in the case of inhalation therapy, the capsule wall does not only serve to “package” the formulation, as its composition and internal surface properties can affect aerosolization and thus the effectiveness of the therapy. For example, excessive adhesion between the capsule wall and the particles of the DPI formulation (this may be due to the static nature of the capsule wall and the roughness of the inner surface) may result in more drug particles remaining in the capsule after inhalation [18,22]. Thus, DPI powder particles can be more difficult to aerosolize and, in carrier-based systems, can also adversely affect the dispersion of the micronized drug from large carrier particles [23]. It should be noted that the properties of DPI capsules may also play a role in the stability of DPI powders, as their residual solvent content (RSC) can affect the structure of formulations (in the case of being amorphous), morphology, density, and interparticle interactions (between drug–drug and/or drug–carrier particles), which also affect the aerosolization and dispersion of the formulations. The stability of the DPI capsules and the increase in fragility over time may also modify the aerodynamics of the powders during inhalation. As a result of the factors listed above, the mass median aerodynamic diameter (MMAD) of the samples may increase and greater deposition is expected in the upper airways, so fine particle fraction (FPF) may be smaller than expected if using DPI capsules improved properties [24,25].

For DPI capsules, three main types can be distinguished. First of all, the use of gelatin (GEL) capsules is widespread, which is still one of the most common type of capsule in capsule-based inhalers on the market, e.g., in Onbrez^®^ Breezhaler^®^ (Novartis International AG, Basel, Switzerland) [26]. However, it should be mentioned that it is incompatible with certain active ingredients (e.g., hydrolyzing agents) and the relatively high RSC involves a risk, since based on experience, it becomes brittle below 10% [16]. The next step was the development of gelatin-PEG (GEL-PEG) capsules. Indeed, their use is not widespread—in a few marketed formulations such capsules they can be found, e.g., in SPIRIVA^®^ HandiHaler^®^ (Boehringer Ingelheim, Ingelheim, Germany)—but for these capsules, the optimal RSC is already lower (10–12%), so they are less exposed to fragmentation than GEL capsules [26]. Another line is hydroxypropyl methylcellulose (HPMC) DPI capsules, e.g., in TOBI^TM^ Podhaler^TM^ (Novartis International AG, Basel, Switzerland), which are prepared using a gelling agent and a network promoter. These capsules are chemically inert, resulting in incompatibility with few materials. Moreover, they have much less optimal RSC (about 3–7%) than the two capsule types detailed earlier, so the risk of fragmentation is even less with this type of DPI capsule [16]. Capsules made from the above-mentioned materials are manufactured/marketed as a separate portfolio for inhalation, in the development of which manufacturers have recently placed increasing emphasis on reducing the static charge of the capsule wall and the adhesion between the powder particles and the capsule wall. Furthermore, it is also important for these capsules to respond well to activation mechanisms such as punching and cutting and to be subject to more stringent microbiological requirements than orally administered capsules [27,28,29].

In the present work, we aimed to investigate the six-month stability test of carrier-free and novel combined formulated DPI microcomposites containing ciprofloxacin hydrochloride (CIP) based on ICH guidelines in three different DPI capsule types (GEL, GEL-PEG, HPMC) and to compare the stability of these two formulations under given conditions. Two of our previously published communications provide the background for this study. In the prior article, results/findings related to the development of the above-mentioned formulations are found [30], while in the second article, stability test results of the same samples were reported at the conditions of 25 ± 2 °C with 50 ± 5% RH (room conditions), stored in open containers for one month [31]. GEL capsules were used in both cases. In our current work, as a novelty, we would like to present a comprehensive approach to the importance of final pharmaceutical dosage form development for the above-mentioned CIP containing samples. Focusing on the stability of each DPI capsule type used and their impact on the stability and in vitro aerodynamic properties of DPI formulations under given conditions. The same formulation may exhibit different stability and thus aerodynamic properties in different DPI capsule types.

## 2. Materials and Methods

### 2.1. Materials

Micronized ciprofloxacin hydrochloride (μCIP) (D (0.5): 5.09 μm) as a fluoroquinolone antibiotic active ingredient was applied and donated by Teva Pharmaceutical Works Ltd. (Debrecen, Hungary). Lactose monohydrate, Inhalac^®^ 70 (IH 70) (D (0.5): 215.00 μm) was gifted by MEGGLE Group (Wasserburg, Germany) and utilized as a carrier. Magnesium stearate (MgSt) (D (0.5): 6.92 μm) was used to treat the surface of IH 70 [32], which was supplied by Sigma-Aldrich (Budapest, Hungary). Sodium stearate (NaSt) (Alfa Aesar, Heysham, United Kingdom) was used as an excipient in the co-spray-drying process. The Coni-Snap^®^ hard GEL (Capsugel^®^/Lonza Pharma & Biotech, Basel, Switzerland), Ezeefit™ GEL-PEG (ACG-Associated Capsules Pvt. Ltd., Mumbai, India) and Ezeeflo™ HPMC (ACG-Associated Capsules Pvt. Ltd., Mumbai, India) capsules were used to store DPI formulations during the stability test.

### 2.2. Methods

#### 2.2.1. Preparation of the Samples

For the six-month-long stability test, we again prepared the formulations which had been investigated in our previous work [30]. The CIP_0.5NaSt_spd microcomposite was produced as a carrier-free DPI system, which was named formulation (1). Furthermore, formulation (2) was the novel combined formulated microcomposite. The former was made with co-spray-drying from a solution of CIP and NaSt. Firstly, the 1.5 *w*/*v* % aqueous solution applying CIP and the ethanolic solution containing 0.0175 *w*/*v* % NaSt were prepared at 30 °C. Then, the two above-mentioned solutions were blended in a ratio of 70:30. Büchi B-191 equipment (Mini Spray Dryer, Büchi Labortechnik AG, Flawil, Switzerland) was utilized for the co-spray-drying process with the following parameters: inlet heating temperature, 130 °C, outlet heating temperature, 78 °C, aspirator capacity, 75%, pressured airflow, 600 L/min, feed pump rate, 5%. So, formulation (1) contained 99.5 *w*/*w* % of drug and 0.5 *w*/*w* % of NaSt. Formulation (2) was the combination of formulation (1) and the surface treated carrier (Figure 1). The surface treatment of IH 70 carrier was performed with 2.0 *w*/*w* % of MgSt [33,34] with Turbula blending (Turbula System Schatz; Willy A. Bachofen AG Maschinenfabrik, Basel, Switzerland) for 4 h [32]. Then, formulation (1) was mixed with a surface modified carrier in the mass ratio of 1:10 [35] with a Turbula blender at 60 rpm for 30 min [36]. Then, knowing their exact drug content, the appropriate amount of the two prepared formulations—see in Section 2.2.2—was filled into GEL, GEL-PEG and HPMC capsules and then blistered, considering that the applied inhalation dose of CIP is 10 mg, which corresponds to ten percent of the oral dose of CIP [37]. As a result, the six samples shown in Table 1 were obtained from the two produced formulations.

#### 2.2.2. Homogeneity and Drug Content Test

After the preparation of formulation (2), homogeneity and drug content investigations were carried out for this microcomposite due to the application of blending actions. The drug content was also tested for formulation (1). The United States Pharmacopeia (USP) required that the tests must be carried out with DPI dosage units [38] taken from ten random places [39]. These were dissolved in distilled water, and the CIP content was calculated with a UV/VIS spectrophotometer (ATIUNICAM UV/VIS Spectrophotometer, Cambridge, UK) at a wavelength of 276 nm. The linearity of CIP in this medium at the above-mentioned wavelength was determined in advance. The linearity of the calibration curve was y = 0.0736x. The unit of the slope was mL/µg.

#### 2.2.3. Investigation of the Stability of the Formulations and the Capsules

Stability tests were performed in a Binder KBF 240 (Binder GmbH Tuttlingen, Germany) constant-climate chamber. An electronically controlled APT.line™ line preheating chamber and refrigerating system ensured temperature accuracy and reproducibility of the results in the temperature range between 10 and 70 °C and the relative humidity (RH) range between 10 and 80%. The stability test was carried out at 40 ± 2 °C with 75 ± 5% RH based on the ICH guideline. The duration of storage of the blistered formulations in different capsule types (six samples) was six months. Sampling was implemented after one month, three months and six months. Under the same conditions, the applied capsule types were stored empty blistered for six months for testing.

#### 2.2.4. Light Microscopic Examination

The shape and area of the holes formed by punching the capsules were recorded with a Leica image analyzer (Leica Q500MC, LEICA Cambridge Ltd., Cambridge, UK) at 4× magnification. Ten replicates per capsule type were performed each time.

#### 2.2.5. Thermoanalytical Test

The Mettler Toledo STAR^e^ (Mettler Inc., Schwerzenbach, Switzerland) was used to determine the RSC of capsule wall types and DPI powders. For thermogravimetry measurements, 3–5 mg of sample per capsule was weighed into 40 μL aluminum crucibles, and the temperature dependence of the mass change of the samples was observed between 25–350 °C at a heating rate of 10 °C/min under nitrogen gas flow. The weight loss up to 110 °C was due to the water leaving the sample.

#### 2.2.6. X-ray Powder Diffraction (XRPD)

The XRPD diffractograms—the raw CIP, NaSt, and the carrier-free formulation during the stability test in the different DPI capsule types—were determined by a BRUKER D8 Advance X-ray powder diffractometer (Bruker AXS GmbH, Karlsruhe, Germany) with Cu K λI radiation (λ = 1.5406 Å) and a VÅNTEC-1 detector. The powders were scanned at 40 kV and 40 mA, with an angular range of 3° to 40° 2θ, at a step time of 0.1 s and a step size of 0.01°.

#### 2.2.7. Particle Size Distribution

Laser diffraction (Malvern Mastersizer Scirocco 2000, Malvern Instruments Ltd., Worcestershire, UK) was applied to determine the particle size distribution of the microcomposites. Approximately 0.5 g of the sample was placed into a feeder tray. The dry analysis method was used, so the air was the dispersion medium for the examined particles. The dispersion air pressure was set to 2.0 bars to determine whether particle attrition had occurred. Three parallel investigations were performed. The D (0.1), D (0.5), and D (0.9) values were determined after the measurements as particle size distribution.

#### 2.2.8. Scanning Electron Microscopy (SEM)

The examination of the morphology of the DPI microcomposites was carried out by scanning electron microscopy (SEM) (Hitachi S4700, Hitachi Scientific Ltd., Tokyo, Japan). For the induction of electric conductivity on the surface of the samples, a sputter coater was used (Bio-Rad SC 502, VG Microtech, Uckfield, UK). The air pressure used was 1.3–13.0 MPa. The formulations were coated with gold-palladium (90 s) under an argon atmosphere using a gold sputter module in a high vacuum evaporator.

#### 2.2.9. In Vitro Aerodynamic Investigation

The in vitro aerodynamic behavior of the DPI samples was examined with an Andersen Cascade Impactor (ACI) (Copley Scientific Ltd., Nottingham, UK) because the ACI is authorized for this purpose in the European Pharmacopoeia, the USP, and the Chinese Pharmacopoeia as well [40]. The plates of the ACI were soaked with a Span^®^ 80 and cyclohexane mixture (1:99) and then allowed to dry. A mass flow meter (Flow Meter Model DFM 2000, Copley Scientific Ltd., Nottingham, UK) with a vacuum pump (High-Capacity Pump Model HCP5, Critical Flow Controller Model TPK, Copley Scientific Ltd., Nottingham, UK) were used to set the appropriate flow rate (28.3 ± 1 L/min), which was applied during the in vitro aerodynamic test. During the in vitro test, three capsules [41] from a given sample were used in one measurement and the Breezhaler^®^ (Novartis, Basel, Switzerland) inhaler was utilized. An inhalation time of 4 s was applied twice for each capsule used. After each test, the inhalator, the DPI capsules used, parts of the ACI (the mouthpiece, the throat, the eight plates (0–7), the filter used) were washed with distilled water. The amount of the drug deposited on these items was determined with an ultraviolet-visible spectrophotometer (ATI-UNICAM UV/VIS Spectrophotometer, Cambridge, UK) at a wavelength of 276 nm. The linearity of the API calibration curve in distilled water was y = 0.0736x at 276 nm (unit of the slope: mL/µg). With the above data known, it is possible to calculate the terms which characterize the in vitro aerodynamic properties of the samples: fine particle fraction (FPF), mass median aerodynamic diameter (MMAD), emitted fraction (EF). EF is the percentage of drug detected from the impactor (from the mouthpiece to the filter)—which is equal to the emitted dose (ED)—relative to the total amount of the API recovered [42]. In the KaleidaGraph 4.0 program (Synergy Software, Reading, PA, USA) the cumulative percentage less than the size range versus the effective cut-off diameter (ACI, 28.3 L/min flow rate [40]) was plotted on the log probability scale. If the abscissa data for the ordinate values of 5 µm and 3 µm are known, the mass with a diameter of less than 5 µm and 3 µm can be determined. The percentage ratios of these amounts to ED are FPF < 5 µm and FPF < 3 µm [43]. The expression of FPF < 3 µm is not yet very common in the international literature, [44,45] since in the deep lung, in the subtracheal area, especially the particles below 3 µm are deposited [46]. The mass median aerodynamic diameter (MMAD) is the diameter at which 50% of the particles of an aerosol by mass are larger and 50% are smaller [47]. This is determined as the ordinate value for the 50% abscissa value. It should be emphasized that the number of DPI capsules used per measurement must also be taken into account in the calculations.

#### 2.2.10. Statistical Analyses

Statistical analyses were carried out applying t-test calculations at a significance level of 0.05 and with a one-tailed hypothesis using the Social Science Statistics, which is available online [48]. All described data indicate ± SD of three parallel measurements (*n* = 3).

## 3. Results and Discussion

### 3.1. Blend Uniformity and Drug Content

For DPIs, blending uniformity should be between 85 and 115% according to the USP criterion and the relative standard deviation (SD) for 10 dosage units should be ≤6%. There is also a stricter 90–110% requirement in the industry [38]. The novel combined carrier-based formulation (2) is also in line with the latter as SD < 5% was obtained (94.17 ± 3.34%), so homogeneity can be assumed [49]. Before the start of the stability period, the DPI capsules were filled with powders in the knowledge of specific drug content. In the case of formulation (1), this value was 98.41 ± 1.07%, even in the case of formulation (2) is 8.518 ± 0.302%.

### 3.2. Stability of the Capsules

Based on Table 2, it can be said that GEL and GEL-PEG capsules started to break even after 1 month. This was especially true for GEL capsules, which formed irregularly shaped holes. The edges of the holes dropped on GEL-PEG capsules were also fractured, although these types of capsules became less brittle during the stability test compared to those containing purely GEL, thus further supporting the viability of the use of PEG. In the case of HPMC capsules, no remarkable change was observed in the shape of the perforated area, and as for the tests, the holes remained approximately regular in terms of their flexibility even after 6 months.

The area of the capsule puncture and the degree of fragmentation during punching increased the most overtime for GEL and GEL-PEG capsules, respectively (Table 2). The initial values of the hole areas (Table 3) for these DPI capsules increased more than 1.5 times after 6 months. There was less area increase for HPMC capsules. The RSC of the capsule walls was also determined after 1, 3 and 6 months of the stability test (Table 3). It was found that the RSC of GEL capsules dropped below the optimal range (13–16%) after the first month, and according to the 3-month results, this was also the case for GEL-PEG capsules (optimal range: 10–12%), while for HPMC capsules the measured values remained within the optimal 3–8% range 6 months later.

### 3.3. Residual Solvent Content of the Samples

The RSC of the samples plays an important role in stability, since in the case of increasing values, recrystallization of the amorphous drug particles, and thus also a structural and morphological change, can be expected. Furthermore, it can contribute to the unfavorable change of interparticle interactions, therefore it can affect the aerosolization and dispersion of the particles, and thus also the lung deposition results. Based on the results of the RSC of the samples determined during the stability test (Table 4), it can be stated that, in general, the values of formulation (1) increased more remarkably in all three DPI capsule types than those of formulation (2). In the latter case, the initial RSC value of around 5% corresponds to the value already published for alpha-lactose monohydrate [50], as it is present in almost 90% of the formulation; however, the effect of MgSt moisture resistance is reflected in the values [51]. Furthermore, for both microcomposites, it was observed that the lowest RSC value was measurable in the HPMC capsule after 6 months, which is related to that described in Section 3.2. It was found that this type of DPI capsule had the smallest decrease in RSC during the stability test, thus less moisture could be transferred to DPI powders.

### 3.4. Structural Investigations

By performing the XRPD examination, it became possible to study the structure of the produced samples before storage and at sampling times for the duration of the stability test. If the XRPD pattern of the raw drug and NaSt is known, conclusions can be drawn regarding the stability of the samples, furthermore, in the case of microcomposites, the dominance of the crystalline or amorphous form affects morphology, so in vitro aerodynamic results can be predicted. In the present study, the XRPD diffractograms of the samples of the carrier-free DPI formulation (1) stored in different capsules are illustrated (Figure 2)—since the pattern of the carrier particles in the case of formulation (2) dominates—during the sampling times of the stability study. For the raw CIP; 8.23, 9.25, 19.22, 26.39, and 29.16 2Teta-degree, even for NaSt, 4.0, 6.0 2Teta-degree characteristic peaks were observed, with crystalline property predominating. The fresh formulation (1) clearly has a predominantly amorphous structural property before storage. After one month, there was no remarkable difference between the XRPD diffractograms of the formulation stored in the different capsules, with some recrystallization seen. After six months, it can be seen that the microcomposites stored in HPMC capsules recrystallized less than those stored in GEL and GEL-PEG capsules. This shows that the particles remained more stable or morphologically less variable during storage in HPMC capsules, which predicts a remarkable difference in in vitro aerodynamic results between the different samples stored in the capsule type.

### 3.5. Particle Size Analysis and Scanning Electron Microscopy (SEM) of the Samples

Detection of changes in the particle size distribution of DPI samples during the stability study is essential, along with the study of morphological properties. It is important for the success of inhalation therapy that the average particle size be between 1 and 5 microns (maximum 10 microns), as several studies have highlighted the fact that most individual particles below 1 micron are exhaled [52], while particles above 5 microns are probably deposited in the upper airways. For formulation (1), D (0.1) and D (0.9) also fell within the optimal range mentioned above throughout the 6-month stability study for all three capsule types (Table 5). The results obtained did not differ remarkably between the capsule types, the average particle size increased slightly better in the GEL capsule type compared to the others. In terms of SEM images, they approximately correlated with the results obtained by laser light scattering. As regards morphology, it can be stated that there is a remarkable difference between the samples stored in different capsule types. After 1 month, recrystallization can be detected in the GEL capsule, which correlates well with our previous stability study (performed under different conditions) [31]. The formulation in this type of capsule appears to be increasingly prone to agglomeration as the stability test progresses. In the case of the GEL-PEG capsule type, recrystallization starts later, so the sample remains stable in this. For HPMC capsules, the particles appear to be the most stable after 6 months.

For formulation (2), the six-month stability study showed that, based on morphological and particle size analysis (Table 6), the sample stored in the HPMC capsule type remained the most stable, with the least aggregation or crystallization appearing. In Table 6, the sample-specific values of D (0.1), D (0.5) and D (0.9) are given, from which the above findings for the products can also be made. However, for more accurate analysis, since the samples contained formulation (1) on the IH70_MgSt surface-treated carrier particles, the D (0.5) values of the drug particles and the surface-modified carrier particles were also taken into account using bimodal distribution curves. Based on these, the value of the drug particle D (0.5) increased from 2.28 µm before storage to 6.129 µm when stored in GEL capsules, 3.004 µm in PEG-GEL capsules, and 2.712 µm even in HPMC capsules. In the case of the surface-treated carrier, the following values were determined: in GEL: 189.313 µm; in PEG-GEL: 176.520 µm and in HPMC capsule: 171.635 µm. Thus, the values measured at the samples were refined for specific components, the same tendencies can be established. Furthermore, comparing formulation (1) and the change in the size of the same D (0.5) in the formulation (2), we can see that the change in average size in the novel combined formulated composite was smaller than in the carrier-free samples. Therefore, higher FPF values for in vitro lung deposition are still expected for formulation (2) compared to the formulation (1), which predicts greater stability of the former (in the HPMC capsule type). The results detailed in this Subsection are closely related to changes in the RSC of DPI capsules and powders during the stability study.

### 3.6. In Vitro Aerodynamic Assessment

Based on the RSC, structure and particle size analysis as well as the SEM images, it can be said that the formulations stored in HPMC capsules (1, 2) remained the most stable considering the physical properties. For both formulations, in vitro aerodynamic tests performed (Table 7 and Table 8) before storage show that the capsule types did not affect FPF values, in both cases the initial FPF values of samples 3-3 were nearly identical. The MMAD values at each measurement point correlated with FPF values over the entire study period. For EF, the initial values showed that in case (1) the drug dripped out of the HPMC capsule better, in case (2) it drifted easily out of all capsule types due to the nature of the formulation. Regarding the FPF values of the 6-month stability study, it can be stated that both formulations tested had the lowest results in the GEL capsules, this was followed by the results of GEL-PEG capsules, and the FPF values decreased the least when using HPMC capsules. The EF values were also the most favorable after 6 months for HPMC capsules, and for sample (1), using this capsule only, the sample meets the prescribed range of 85–115%. For EF, it was also observed that the SD was higher for samples 1_GEL and 2_GEL compared to the other samples. This is explained by the results presented in Section 3.2, i.e., the area of capsule puncture measured in the case of the GEL capsule and its SD, and the SEM images of GEL capsules also serve as support.

The results of formulations (1) and (2), when considered, correlate with the results of our previous publications for prestorage values. It can be stated that the novel combined carrier-based formulation (2) achieved better in vitro aerodynamic results under the aforementioned storage conditions—in all capsule types—than the carrier-free formulation (1), which corresponds to the results of the 1-month stability test previously performed at room temperature [31].

## 4. Conclusions

In this study we introduced the importance of final formulation-development by studying the effect of capsule types on the stability and aerodynamic properties of DPI. The same formulation have different stability and thus aerodynamic properties in different DPI capsule types. The RSC and light microscopic results of the DPI capsules supported the claim that GEL and GEL-PEG-type capsules begin to break when the RSC falls below the optimal range. Due to their fragmentation, the resulting holes became irregularly shaped and large. Although more formulations came out of these larger, irregularly shaped holes, resulting in increased EF values, the deaggregation of the particles was less efficient, which in turn reduced FPF values. However, HPMC capsules retained their elasticity after 6 months, pieces of the capsule wall did not break during punching, and the holes remained in regular shape. RSC and XRPD analysis confirmed, and the SEM images also showed that DPI powders stored in GEL and GEL-PEG capsules formed irregularly shaped particles during the stability study due to the onset of recrystallization (it is assumed that moisture was transferred to DPI powders). The altered habit was aerodynamically disadvantageous, which may have been one of the reasons for the decrease in FPF values. The morphological change was least observed with the formulations stored in HPMC capsules, and FPF values decreased to a lesser extent. Overall, initial, almost identical aerosolization values after 6 months were the most favorable for HPMC capsules for both investigated DPI formulations. This was probably due to the RSC of the capsules, the size and shape of the perforated area, and the altered habit of the DPI powder. The results of the novel combined formulated composite were more favorable after the stability test than those of the carrier-free formulation for all DPI capsule types.

Thus, it may be worthwhile focusing on testing DPI formulations in different capsules during pulmonary dosage form development, as the same formulation may have different stability and thus aerodynamic properties in different DPI capsule types. The prepared DPI formulation of a carrier-free and novel combined carrier-based systems using CIP could present an effective new possibility in the therapy of lung diseases (direct and indirect treatment of pathophysiological processes such as cystic fibrosis and chronic bronchitis) instead of the per os applied antibiotic formulation.

## Figures and Tables

**Figure 1 pharmaceutics-13-00689-f001:**
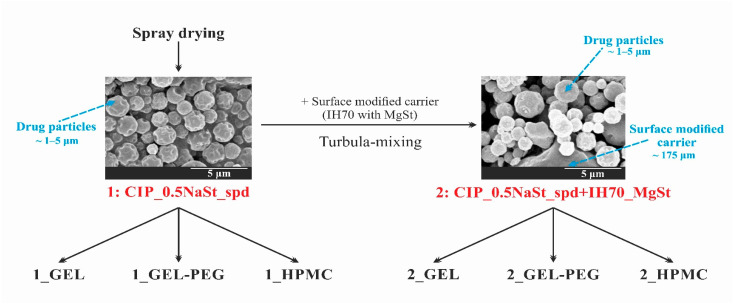
Schematic overview of the preparation.

**Figure 2 pharmaceutics-13-00689-f002:**
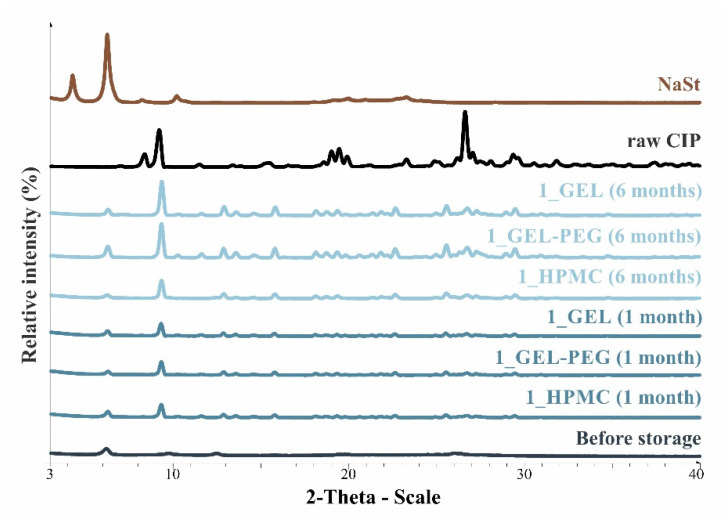
XRPD patterns of raw CIP and formulation 1 during the stability test.

**Table 1 pharmaceutics-13-00689-t001:** Details of the components of the samples.

Samples	Compositions of the DPI Formulations				Applied DPI Capsule Types		
	CIP (*w*/*w* %)	NaSt (*w*/*w* %)	IH 70 (*w*/*w* %)	MgSt (*w*/*w* %)	GEL	GEL-PEG	HPMC
1_GEL	99.50	0.500	–	–	+	–	–
1_GEL-PEG	99.50	0.500	–	–	–	+	–
1_HPMC	99.50	0.500	–	–	–	–	+
2_GEL	9.045	0.045	88.91	2.000	+	–	–
2_GEL-PEG	9.045	0.045	88.91	2.000	–	+	–
2_HPMC	9.045	0.045	88.91	2.000	–	–	+

**Table 2 pharmaceutics-13-00689-t002:** Light microscopic images of the punctured ends of the applied DPI capsules.

Capsule Type	Before Storage	1 Month	3 Months	6 Months
GEL	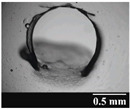	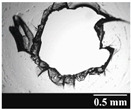	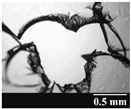	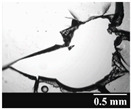
GEL-PEG	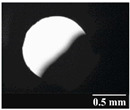	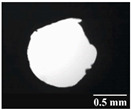	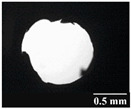	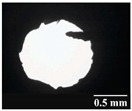
HPMC	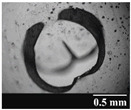	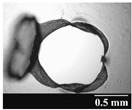	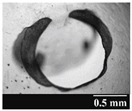	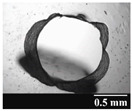

**Table 3 pharmaceutics-13-00689-t003:** RSC of capsule walls and areas of holes formed during punching.

Capsule Type	Time	RSC (%)	Area of Capsule Puncture (mm^2^)
GEL	Before storage	15.26 ± 0.18	0.60 ± 0.16
	1 month	10.31 ± 0.21	0.74 ± 0.11
	3 months	7.23 ± 0.28	1.01 ± 0.28
	6 months	6.68 ± 0.12	1.14 ± 0.38
GEL-PEG	Before storage	11.87 ± 0.09	0.54 ± 0.10
	1 month	10.68 ± 0.32	0.84 ± 0.12
	3 months	8.74 ± 0.15	0.89 ± 0.14
	6 months	7.12 ± 0.12	0.92 ± 0.07
HPMC	Before storage	5.98 ± 0.11	0.79 ± 0.05
	1 month	5.45 ± 0.09	0.79 ± 0.04
	3 months	4.84 ± 0.13	0.86 ± 0.08
	6 months	4.62 ± 0.02	0.88 ± 0.03

**Table 4 pharmaceutics-13-00689-t004:** RSC values of DPI powders during the stability test.

Samples	RSC (%)			
	Before Storage	1 Month	3 Months	6 Months
Formulation (1)	3.76 ± 0.07	—	—	—
Formulation (2)	4.61 ± 0.12	—	—	—
1_GEL	—	3.99 ± 0.06	4.62 ± 0.08	5.21 ± 0.08
1_GEL-PEG	—	3.92 ± 0.03	4.48 ± 0.06	5.03 ± 0.09
1_HPMC	—	3.85 ± 0.10	4.26 ± 0.13	4.72 ± 0.04
2_GEL	—	4.93 ± 0.11	5.13 ± 0.09	5.64 ± 0.13
2_GEL-PEG	—	4.85 ± 0.06	5.04 ± 0.03	5.45 ± 0.06
2_HPMC	—	4.76 ± 0.07	4.91 ± 0.06	5.16 ± 0.04

**Table 5 pharmaceutics-13-00689-t005:** Particle size distribution and morphology of the carrier-free samples during the stability test.

Formulation				1					
Before storage				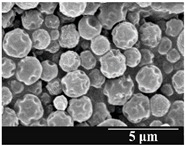					
				D (0.1) (µm)	D (0.5) (µm)	D (0.9) (µm)			
				1.167 ± 0.07	2.167 ± 0.10	3.715 ± 0.12			
Samples	1_GEL			1_GEL-PEG			1_HPMC		
1 month	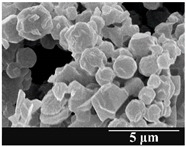			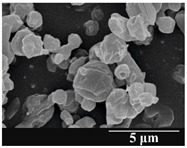			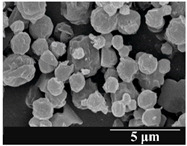		
	D (0.1) (µm)	D (0.5) (µm)	D (0.9) (µm)	D (0.1) (µm)	D (0.5) (µm)	D (0.9) (µm)	D (0.1) (µm)	D (0.5) (µm)	D (0.9) (µm)
	1.232 ± 0.08	2.264 ± 0.11	4.022 ± 0.11	1.313 ± 0.02	2.231 ± 0.03	3.788 ± 0.15	1.247 ± 0.08	2.230 ± 0.10	3.804 ± 0.13
3 months	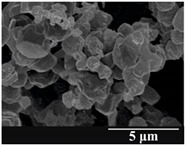			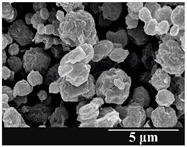			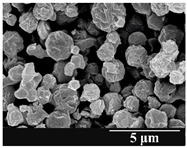		
	D (0.1) (µm)	D (0.5) (µm)	D (0.9) (µm)	D (0.1) (µm)	D (0.5) (µm)	D (0.9) (µm)	D (0.1) (µm)	D (0.5) (µm)	D (0.9) (µm)
	1.244 ± 0.03	2.828 ± 0.14	4.468 ± 0.13	1.210 ± 0.03	2.769 ± 0.12	4.662 ± 0.18	1.442 ± 0.06	2.711 ± 0.07	4.454 ± 0.14
6 months	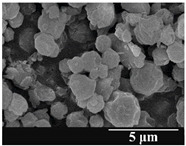			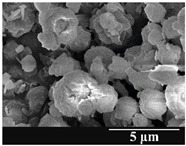			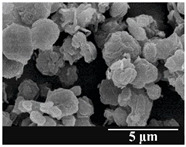		
	D (0.1) (µm)	D (0.5) (µm)	D (0.9) (µm)	D (0.1) (µm)	D (0.5) (µm)	D (0.9) (µm)	D (0.1) (µm)	D (0.5) (µm)	D (0.9) (µm)
	1.329 ± 0.06	3.615 ± 0.17	7.929 ± 0.21	1.262 ± 0.11	3.452 ± 0.05	6.731 ± 0.13	1.358 ± 0.02	3.460 ± 0.15	6.371 ± 0.09

**Table 6 pharmaceutics-13-00689-t006:** Particle size distribution and morphology of the novel combined carrier-based samples during the stability test.

Formulation				2					
Before storage				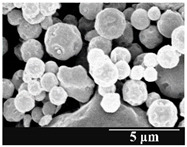					
				D (0.1) (µm)	D (0.5) (µm)	D (0.9)(µm)			
				3.675 ± 0.12	130.459 ± 0.18	235.25 ± 1.15			
Samples	2_GEL			2_GEL-PEG			2_HPMC		
1 month	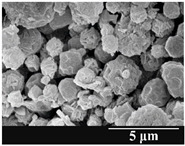			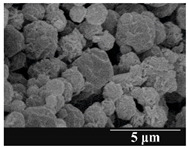			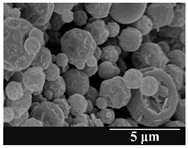		
	D (0.1) (µm)	D (0.5) (µm)	D (0.9) (µm)	D (0.1) (µm)	D (0.5) (µm)	D (0.9) (µm)	D (0.1) (µm)	D (0.5) (µm)	D (0.9) (µm)
	14.389 ± 0.23	160.591 ± 1.09	317.334 ± 1.76	9.128 ± 0.21	158.867 ± 0.54	280.981 ± 1.18	3.913 ± 0.11	155.349 ± 0.71	273.114 ± 1.42
3 months	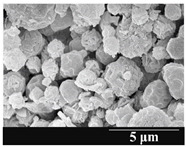			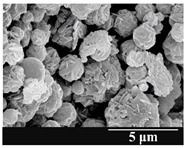			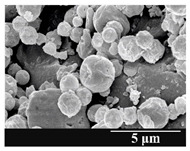		
	D (0.1) (µm)	D (0.5) (µm)	D (0.9) (µm)	D (0.1) (µm)	D (0.5) (µm)	D (0.9) (µm)	D (0.1) (µm)	D (0.5) (µm)	D (0.9) (µm)
	29.426 ± 0.19	168.583 ± 0.71	305.176 ± 1.81	22.836 ± 0.13	166.571 ± 0.86	303.715 ± 0.96	22.315 ± 0.31	164.727 ± 0.38	291.028 ± 1.23
6 months	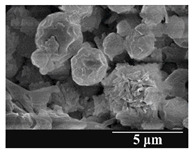			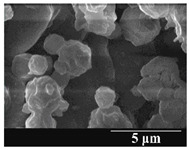			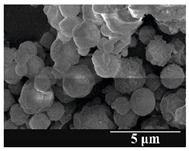		
	D (0.1) (µm)	D (0.5) (µm)	D (0.9) (µm)	D (0.1) (µm)	D (0.5) (µm)	D (0.9) (µm)	D (0.1) (µm)	D (0.5) (µm)	D (0.9) (µm)
	27.381 ± 0.08	172.772 ± 0.36	331.195 ± 1.39	29.003 ± 0.15	170.503 ± 0.37	328.693 ± 1.41	26.122 ± 0.18	168.635 ± 0.89	305.315 ± 1.72

**Table 7 pharmaceutics-13-00689-t007:** Aerodynamic properties of the carrier-free formulations.

Samples	Time	FPF (%) <5 μm	MMAD (μm)	EF (%)
1_GEL	Before storage	53.42 ± 1.23	3.98 ± 0.15	77.04 ± 1.03
	1 month	31.87 ± 0.11	4.43 ± 0.14	85.75 ± 0.16
	3 months	29.94 ± 0.25	4.86 ± 0.17	86.14 ± 0.81
	6 months	28.83 ± 0.65	5.02 ± 0.22	87.70 ± 0.64
1_GEL-PEG	Before storage	54.13 ± 0.89	3.81 ± 0.06	72.72 ± 0.76
	1 month	42.25 ± 0.38	4.31 ± 0.21	86.54 ± 0.54
	3 months	36.31 ± 0.43	4.62 ± 0.15	86.85 ± 0.85
	6 months	31.67 ± 0.07	4.93 ± 0.12	87.80 ± 0.73
1_HPMC	Before storage	53.97 ± 1.08	3.78 ± 0.26	86.44 ± 0.99
	1 month	44.71 ± 0.94	4.16 ± 0.14	86.96 ± 0.36
	3 months	39.18 ± 0.27	4.32 ± 0.08	87.55 ± 0.49
	6 months	38.59 ± 0.44	4.40 ± 0.11	90.16 ± 0.34

**Table 8 pharmaceutics-13-00689-t008:** Aerodynamic properties of the novel combined carrier-based samples.

Samples	Time	FPF (%) <5 μm	MMAD (μm)	EF (%)
2_GEL	Before storage	62.91 ± 1.02	3.51 ± 0.09	90.31 ± 0.95
	1 month	43.89 ± 1.28	3.84 ± 0.13	91.15 ± 0.12
	3 months	35.03 ± 0.23	3.93 ± 0.07	91.27 ± 0.36
	6 months	31.71 ± 0.64	4.10 ± 0.16	92.89 ± 0.41
2_GEL-PEG	Before storage	62.53 ± 0.48	3.45 ± 0.12	90.21 ± 0.83
	1 month	46.11 ± 1.32	3.72 ± 0.05	89.75 ± 0.45
	3 months	38.66 ± 0.96	3.87 ± 0.09	91.39 ± 0.21
	6 months	36.26 ± 0.39	4.03 ± 0.13	92.56 ± 0.66
2_HPMC	Before storage	63.15 ± 0.41	3.47 ± 0.08	89.55 ± 0.26
	1 month	53.29 ± 0.72	3.68 ± 0.21	91.42 ± 0.52
	3 months	45.23 ± 1.12	3.84 ± 0.04	94.39 ± 0.74
	6 months	43.40 ± 0.57	3.91 ± 0.15	96.98 ± 0.63

## Data Availability

The data presented in this study are available on request from the corresponding author.

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
