# Peer review of "Stability and In Vitro Aerodynamic Studies of Inhalation Powders Containing Ciprofloxacin Hydrochloride Applying Different DPI Capsule Types"

_pharmaceutics, 2021, doi:10.3390/pharmaceutics13050689_

Round 1
Reviewer 1 Report
The manuscript is very interesting, generally clear and the methodology used is appropriated. The introduction is properly written, and the results are well presented, supporting the conclusions.
Reviewer has several minor suggestions:
1) In line 246, Results and Discussion seem to be better than only Results.
2) In 3.1. Blend Uniformity and Drug Content, the obtained values of them in formulation (2) should be showed.
3) In table 6, the samples names are not correct. Please check and revise them.
4) In line 374, FFF seems to be wrong. FPF is correct.
Author Response
Comments and Suggestions for Authors:
The manuscript is very interesting, generally clear and the methodology used is appropriated. The introduction is properly written, and the results are well presented, supporting the conclusions.
Thank you very much for your positive feedback on our work.
Reviewer has several minor suggestions:
- In line 246, Results and Discussion seem to be better than only Results.
Thank you for your comment. It has been modified in the manuscript.
2) In 3.1. Blend Uniformity and Drug Content, the obtained values of them in formulation (2) should be showed.
Thank you very much for your suggestion. The 3.1. subsection has been supplemented as follows:
3.1. Blend Uniformity and Drug Content
For DPIs, blending uniformity should be between 85 and 115% according to the USP criterion and the relative standard deviation (SD) for 10 dosage units should be ≤6%. There is also a stricter 90-110% requirement in the industry [38]. The novel combined carrier-based formulation (2) is also in line with the latter as SD <5% was obtained (94.17 ± 3.34%), so homogeneity can be assumed [49]. Before the start of the stability period, the DPI capsules were filled with powders in the knowledge of specific drug content. In the case of formulation (1), this value was 98.41 ± 1.07%, even in the case of formulation (2) is 8.518 ± 0.302%.
3) In table 6, the samples names are not correct. Please check and revise them.
Thank you for your remark. A typo occurred, we corrected.
4) In line 374, FFF seems to be wrong. FPF is correct.
Unfortunately, it was typing error. We corrected.
Szeged, 26. 04. 2021.
Prof. Dr. Rita Ambrus

Reviewer 2 Report
Dear authors,
I consider the manuscript: "Importance of capsule-type on the stability and in vitro aero-solization of dry powder inhalation formulations" very well written, clear and consistent.
The study was well planned and executed. The results are consistent and important for the development of a novel pharmaceutical inhalation formulation. The results were well discussed and the bibliographical references are relevant and current. I have only one observation: in table 6 the samples aren't write correctly; the authors must correct this table.
Finally, I believe the manuscript is of interest to the readers of the journal.
Best regards
Annarita Stringaro
Author Response
Comments and Suggestions for Authors:
Dear authors,
I consider the manuscript: "Importance of capsule-type on the stability and in vitro aero-solization of dry powder inhalation formulations" very well written, clear and consistent.
The study was well planned and executed. The results are consistent and important for the development of a novel pharmaceutical inhalation formulation. The results were well discussed and the bibliographical references are relevant and current. I have only one observation: in table 6 the samples aren't write correctly; the authors must correct this table.
Finally, I believe the manuscript is of interest to the readers of the journal.
Dear reviewer,
Thank you very much for the positive feedback on our manuscript. It is true that a typing error occurred in table 6, this has been modified (the correction is highlighted in green because another reviewer also had this comment).
Szeged, 26. 04. 2021.
Prof. Dr. Rita Ambrus

Reviewer 3 Report
For the reader, the main interest of this paper actually is on the development of the formulation reported as formulation (2) that are novel combined carrier-based samples and not on the importance of capsule. In fact, according to the reviewer, the issues associated with the use of gelatin capsule when dealing with dry powder for inhalation are well-known. For this reason, HPMC capsules are now the reference standard in the field of dry powders for inhalaiton. Therefore, I believe that the results of this study, even though the methods are clearly described and the results are clearly presented, were expected so the scientific soundness and interest for the readers are low. I believe the paper does not bring novelty and is not worthy of being published in Pharmaceutics.
Author Response
Comments and Suggestions for Authors:
For the reader, the main interest of this paper actually is on the development of the formulation reported as formulation (2) that are novel combined carrier-based samples and not on the importance of capsule. In fact, according to the reviewer, the issues associated with the use of gelatin capsule when dealing with dry powder for inhalation are well-known. For this reason, HPMC capsules are now the reference standard in the field of dry powders for inhalaiton. Therefore, I believe that the results of this study, even though the methods are clearly described and the results are clearly presented, were expected so the scientific soundness and interest for the readers are low. I believe the paper does not bring novelty and is not worthy of being published in Pharmaceutics.
Thanks for the comment. It is basically true, that the application of gelatin and HPMC capsules in DPI fromulation is well known. During our work we would introduce the importance of final formulation-development by studying the effect of capsule types on the stability and aerodynamic properties. The same formulation may have different stability and thus aerodynamic properties in different DPI capsule types. By the marketed formulations both types of capsules are used, as you can see in the attached table. The novelty of our work is the preparation and DPI formulation of a carrier-free and novel combined carrier-based systems using ciprofloxacine HCl (CIP), therefore present an effective new possibility in the therapy of lung diseases (direct and indirect treatment of pathophysiological processes such as cystic fibrosis and chronic bronchitis) instead of the per os applied antibiotic formulation.
DPI device |
Company |
Capsule wall |
|
Aerohaler™ |
Boehringer-Ingelheim |
gelatin |
|
Aerolizer™ |
Novartis |
gelatin |
|
ARCUS TM |
Acorda Therapeutics |
HPMC |
|
Axahaler® |
Sager Pharma |
HPMC |
|
Onbrez |
Breezhaler® |
Novartis |
gelatin |
Seebri |
HPMC |
||
Ultibro |
HPMC |
||
FlowCaps® |
Hovione |
HPMC |
|
Handihaler® |
Boehringer Ingelheim |
gelatin-PEG |
|
Podhaler™ |
Novartis |
HPMC |
|
PowdAir® Plus |
Hovione |
gelatin; HPMC |
|
Revolizer |
Cipla |
gelatin; HPMC |
|
Rotahaler® |
GSK |
gelatin |
|
Spinhaler® |
Fission/Aventis |
gelatin |
|
Turbospin® |
PH&T |
gelatin-PEG |
|
Twister® |
Aptar Pharma |
gelatin; HPMC |
|
Zonda® inhaler |
TEVA |
HPMC |
References:
[1] S. Reddy, PAST AND PRESENT TRENDS OF DRY POWDER INHALER DEVICES: A REVIEW, Journal of Drug Delivery and Therapeutics. 4 (2014). https://doi.org/10.22270/jddt.v4i1.726.
[2] K. Berkenfeld, A. Lamprecht, J.T. McConville, Devices for Dry Powder Drug Delivery to the Lung, AAPS PharmSciTech. 16 (2015) 479–490. https://doi.org/10.1208/s12249-015-0317-x.
[3] PHARMINDEX Online. (n.d.). https://www.pharmindex-online.hu/gyogyszerkereso/kereses (accessed March 30, 2019).
[4] Review of Dry Powder Inhaler Devices | American Pharmaceutical Review - The Review of American Pharmaceutical Business & Technology, (n.d.). https://www.americanpharmaceuticalreview.com/Featured-Articles/185892-Review-of-Dry-Powder-Inhaler-Devices/ (accessed March 30, 2019).
[5] F. Martinelli, A.G. Balducci, A. Rossi, F. Sonvico, P. Colombo, F. Buttini, “Pierce and inhale” design in capsule based dry powder inhalers: Effect of capsule piercing and motion on aerodynamic performance of drugs, International Journal of Pharmaceutics. 487 (2015) 197–204. https://doi.org/10.1016/j.ijpharm.2015.04.003.
[6] D.A. Dean, E.R. Evans, I.H. Hall, Pharmaceutical Packaging Technology., 2000. http://ebookcentral.proquest.com/lib/qut/detail.action?docID=240114 (accessed March 31, 2019).
[7] Dry Powder Inhalers - Twister | Aptar Pharma, (n.d.). https://pharma.aptar.com/en-us/dispensing-solutions/twisterr.html (accessed February 17, 2020).
Szeged, 26. 04. 2021.
Prof. Dr. Rita Ambrus

Round 2
Reviewer 3 Report
I believe the authors should modify the focus of their work on the development of the new formulation that is formulation (2) so that the work is of interest to the readers. Therefore, the title of their manuscript should be modfied and the paper should focus on the formulationdevelopment and characterization rather than on the effect of the capsule type. Then, the effect of capsule can also be reported but should not be the main topic of the paper. The manuscript should be deeply restructured.
Author Response
Reply to Reviewer 3. comments (Round 2.)
Thank you for your comments. Below are the answers to your suggestions whereas modifications on the manuscript are colored by blue.
Comments and Suggestions for Authors:
I believe the authors should modify the focus of their work on the development of the new formulation that is formulation (2) so that the work is of interest to the readers. Therefore, the title of their manuscript should be modfied and the paper should focus on the formulationdevelopment and characterization rather than on the effect of the capsule type. Then, the effect of capsule can also be reported but should not be the main topic of the paper. The manuscript should be deeply restructured.
Dear Editors and Reviewer 3,
At the editor's request, we have also modified the manuscript to take into account your comments on the focus of the paper, which we hope will clarify the manuscript's goal and presentation of its results.We have also reworded the title to be more specific.
The background to this publication is two of our previous articles, which are:
Ambrus, R.; Benke, E.; Farkas, Á.; Balásházy, I.; Szabó-Révész, P. Novel Dry Powder Inhaler Formulation Containing Antibiotic Using Combined Technology to Improve Aerodynamic Properties. Eur. J. Pharm. Sci. 2018, 123, 20–27, doi:10.1016/j.ejps.2018.07.030.
In this communication, the development results of a novel combined carrier-based formulation (in this publication formulation (2)) are presented, with the following brief findings:
„The purpose of our research work was the development of a novel antibiotic-containing DPI including the advantages of carrier-based and carrier-free formulations. The samples containing ciprofloxacin hydrochloride (CIP), lipophilic additive (NaSt), surface modifier (MgSt) and inhalable carrier (IH70) were characterized. Based on the comparison of the results we concluded that cohesive-adhesive balance could be modified with the particle engineering of the drug and the presence of the excipients. Therefore we offer a new possibility in DPI formulation to improve the in vitro-in silico aerodynamic properties. The in silico deposition results were in line with the in vitro measurements and yielded increased lung doses for the sample prepared by the combination method. The present work demonstrates that the novel procedure for the formulation of CIP DPI offers a more effective therapy for cystic fibrosis with deeper deposition of the drug.”
Benke, E.; Farkas, Á.; Balásházy, I.; Szabó-Révész, P.; Ambrus, R. Stability Test of Novel Combined Formulated Dry Powder Inhalation System Containing Antibiotic: Physical Characterization and in Vitro – in Silico Lung Deposition Results. Drug Dev. Ind. Pharm. 2019, 45, 1369–1378, doi:10.1080/03639045.2019.1620268.
In the following article, stability test results of the same samples were reported at the (room) conditions of 25 ± 2 °C with 50 ± 5% RH in open containers for 1 month. Furthermore, gelatin capsules were used in both the first and second articles, as gelatin capsules are widely available on the market and have been applied in our previous developments.
In the light of these articles, the research and writing of the present submitted manuscript was carried out, where samples CIP_0.5NaSt_spd and CIP_0.5NaSt_spd+IH70_MgSt were tested for stability (blistered) for 6 months in 3 different capsule types according to ICH guidelines. The stability of the formulations can vary with changes in conditions, and it has been shown that formulation (2) shows better results than formulation (1) under these conditions, too. The current communication presents much more comprehensive results than the 1-month stability study carried out at room temperature only where the role of capsules was not reported. Thus, the focus of the publication is on this, not on the formulation results. The role of DPI capsules is a very up to date topic in the development of inhalation formulations [1–4], and there has recently been a strong emphasis on the role of capsules in influencing the aerosolization of formulations. As the communication shows, not without reason. We believe that the manuscript draws attention to the importance of capsule selection and its impact on the efficacy of DPI formulations. Many publications report results using only the gelatin capsule type [5–7].
It may be the result of inappropriate capsule selection that a relatively good formulation does not meet international standards, e.g. in terms of emitted fraction, stability, which are affected by the properties of the capsules. In addition, incompatibility problems may occur with gelatine capsules, which can even be avoided when using HPMC capsules.
References:
- Wauthoz, N.; Hennia, I.; Ecenarro, S.; Amighi, K. Impact of Capsule Type on Aerodynamic Performance of Inhalation Products: A Case Study Using a Formoterol-Lactose Binary or Ternary Blend. International Journal of Pharmaceutics 2018, 553, 47–56, doi:10.1016/j.ijpharm.2018.10.034.
- Capsugel Capsugel® ZephyrTM - Dry-Powder Inhalation Capsule Portfolio Available online: https://www.capsugel.com/biopharmaceutical-products/capsugel-zephyr-dry-powder-inhalation-capsule-portfolio (accessed on 18 March 2021).
- Lavorini, F.; Pistolesi, M.; Usmani, O.S. Recent Advances in Capsule-Based Dry Powder Inhaler Technology. Multidiscip Respir Med 2017, 12, 11, doi:10.1186/s40248-017-0092-5.
- Pinto, J.T.; Wutscher, T.; Stankovic-Brandl, M.; Zellnitz, S.; Biserni, S.; Mercandelli, A.; Kobler, M.; Buttini, F.; Andrade, L.; Daza, V.; et al. Evaluation of the Physico-Mechanical Properties and Electrostatic Charging Behavior of Different Capsule Types for Inhalation Under Distinct Environmental Conditions. AAPS PharmSciTech 2020, 21, 128, doi:10.1208/s12249-020-01676-2.
- Pinto, J.T.; Cachola, I.; F. Pinto, J.; Paudel, A. Understanding Carrier Performance in Low-Dose Dry Powder Inhalation: An In Vitro–In Silico Approach. Pharmaceutics 2021, 13, 297, doi:10.3390/pharmaceutics13030297.
- Faulhammer, E.; Zellnitz, S.; Wutscher, T.; Stranzinger, S.; Zimmer, A.; Paudel, A. Performance Indicators for Carrier-Based DPIs: Carrier Surface Properties for Capsule Filling and API Properties for in Vitro Aerosolisation. International Journal of Pharmaceutics 2018, 536, 326–335, doi:10.1016/j.ijpharm.2017.12.004.
- Varshosaz, J.; Taymouri, S.; Hamishehkar, H.; Vatankhah, R. Development of Dry Powder Inhaler Containing Tadalafil-Loaded PLGA Nanoparticles. Res Pharm Sci. 2017, 12, 222–232, doi:10.4103/1735-5362.207203.
Szeged, 03. 05. 2021.
Prof. Dr. Rita Ambrus
